# Renal Outcome of IgM Nephropathy: A Comparative Prospective Cohort Study

**DOI:** 10.3390/jcm10184191

**Published:** 2021-09-16

**Authors:** Yura Chae, Hye Eun Yoon, Yoon Kyung Chang, Young Soo Kim, Hyung Wook Kim, Bum Soon Choi, Cheol Whee Park, Ho Cheol Song, Young Ok Kim, Eun Sil Koh, Sungjin Chung

**Affiliations:** 1Department of Internal Medicine, College of Medicine, The Catholic University of Korea, Seoul 06591, Korea; 22000658@cmcnu.or.kr (Y.C.); berrynana@catholic.ac.kr (H.E.Y.); racer@catholic.ac.kr (Y.K.C.); dr52916@catholic.ac.kr (Y.S.K.); khw@catholic.ac.kr (H.W.K.); sooncb@catholic.ac.kr (B.S.C.); cheolwhee@hanmail.net (C.W.P.); drsong@catholic.ac.kr (H.C.S.); cmckyo@catholic.ac.kr (Y.O.K.); 2Division of Nephrology, Yeouido St. Mary’s Hospital, The Catholic University of Korea, Seoul 07345, Korea; 3Division of Nephrology, Incheon St. Mary’s Hospital, The Catholic University of Korea, Incheon 22711, Korea; 4Division of Nephrology, Daejeon St. Mary’s Hospital, The Catholic University of Korea, Daejeon 34943, Korea; 5Division of Nephrology, Uijeongbu St. Mary’s Hospital, The Catholic University of Korea, Uijeongbu 11765, Korea; 6Division of Nephrology, St. Vincent’s Hospital, The Catholic University of Korea, Suwon 16247, Korea; 7Division of Nephrology, Eunpyeong St. Mary’s Hospital, The Catholic University of Korea, Seoul 03476, Korea; 8Division of Nephrology, Seoul St. Mary’s Hospital, The Catholic University of Korea, Seoul 06591, Korea; 9Division of Nephrology, Bucheon St. Mary’s Hospital, The Catholic University of Korea, Bucheon 14647, Korea

**Keywords:** IgM nephropathy, kidney biopsy, glomerulonephritis, renal outcome, prognosis

## Abstract

Immunoglobulin M nephropathy (IgMN) is an idiopathic glomerulonephritis characterized by diffuse deposits of IgM in the glomerular mesangium. However, its renal prognosis remains unknown. We compared renal outcomes of IgMN patients with those of patients with minimal change disease (MCD), focal segmental glomerulosclerosis (FSGS), or mesangial proliferative glomerulonephritis (MsPGN) from a prospective observational cohort, with 1791 patients undergoing native kidney biopsy in eight hospitals affiliated with The Catholic University of Korea between December 2014 and October 2020. IgMN had more mesangial proliferation and matrix expansion than MsPGN and more tubular atrophy and interstitial fibrosis than MCD. IgMN patients had decreased eGFR than MCD patients in the earlier follow-up. However, there was no significant difference in urine protein or eGFR among all patients at the last follow-up. When IgMN was divided into three subtypes, patients with FSGS-like IgMN tended to have lower eGFR than those with MCD-like or MsPGN-like IgMN but higher proteinuria than MsPGN-like IgMN without showing a significant difference. The presence of hypertension at the time of kidney biopsy predicted ≥20% decline of eGFR over two years in IgMN patients. Our data indicate that IgMN would have a clinical course and renal prognosis similar to MCD, FSGS, and MsPGN.

## 1. Introduction

Since the first description of Immunoglobulin M (IgM) nephropathy (IgMN) in the 1970s [1,2,3], there has still been controversy about the independence of IgMN in the range of glomerular diseases [3,4]. IgMN is pathologically characterized by diffuse deposits of IgM in the mesangium at immunofluorescence [1,5,6]. Light microscopy has demonstrated various histological pictures, ranging from no glomerular abnormality to mesangial hyperplasia and accumulation of the extracellular mesangial matrix of varying degrees, associated with segmental or global sclerosis of the glomeruli [3,4,5,6,7]. Because of these varying morphological characteristics, IgMN as an independent entity has been questioned [4]. Some investigations have indicated that IgMN resembles minimal change disease (MCD) and focal and segmental glomerulosclerosis (FSGS), while others have argued that IgMN is a transitional form between these two disorders [3,4,5,6,7]. Furthermore, there is still no consensus for the diagnostic criteria of IgMN with respect to the intensity of IgM staining, the presence of other immunoreactants, the degree of mesangial proliferation, or electron microscopic findings [3]. Since the distinctness of this entity has been the subject of debate [8,9], some researchers tend to be reluctant to make reference to the disease. However, it is necessary to be reminded that it takes a long time for immunoglobulin A (IgA) nephropathy to become recognized as a discrete entity [10]. In addition, a consensus on defining characteristics of C3 glomerulopathy was presented only a few years ago [11]. Moreover, a consensus definition on the diagnosis of C1q nephropathy is lacking, and its existence as a distinct clinical disease entity remains controversial [12]. Therefore, there are still arguments about the definitive diagnosis and methods to identify and classify IgM nephropathy before a global and robust consensus is reached.

The reported frequency of IgMN in the literature has varied from 1.8% to 18.5% in native biopsies [3,5,13]. IgMN appears to have different clinical outcomes at different ages. It mainly presents proteinuria or hematuria in young adults or children [3,6]. Proteinuria in IgMN can range from asymptomatic proteinuria to nephrotic syndrome [3]. On renal biopsy, light microscopy (LM) shows varying degrees of mesangial cell proliferation or mesangial sclerosis from minor changes [1,3,7]. A few cases have been reported in the form of crescentic glomerulonephritis (GN) [5,7,14]. When studying immunofluorescence (IF) in kidney biopsy specimens, IgM deposits in the mesangium are observed in a diffuse or granular pattern [1,4]. Although other immunoglobulins other than IgM can be observed minorly, predominant IgM deposition is usually characteristic [3,7]. The presence of complement fragments, such as C3 or C1q, has also been reported in some studies [1,3,6]. An electron microscopy (EM) study may not be essential for the diagnosis of IgMN. Previous studies have demonstrated granular to short linear electron-dense deposits or podocyte foot effacement on EM [3,4].

Renal prognosis varies according to reports [3,7]. Despite the scanty data on long-term outcomes of patients with IgMN, a few studies have suggested that IgMN has a clinical course between MCD and FSGS [3,4]. However, few studies have compared the prognosis between IgMN and other glomerular diseases. Thus, the objective of this study was to investigate whether there might be a significant difference in renal outcome between biopsy-proven IgMN and other representative glomerular diseases, such as MCD, FSGS, and IF-negative nonspecific mesangial proliferative glomerulonephritis (MsPGN). In addition, we evaluated whether there could be differences in clinical and pathological characteristics according to variable LM findings of IgMN, such as MCD-like, FSGS-like, and MsPGN-like IgMN, and what factors could influence the renal outcome of IgMN.

## 2. Materials and Methods

### 2.1. Definition

The diagnostic criteria we adopted for IgMN were stricter in this study. IgMN was defined and classified only when only IgM was positive on the IF study (Ig M positivity 1+ to 3+) without deposition with any other type of immunoglobulin. Among subtypes of IgMN, FSGS-like IgMN meant a case in which glomerulosclerosis was shown like FSGS on its LM finding. MCD-like IgMN was defined as a case with proteinuria suitable for nephrotic syndrome, with an LM finding compatible with MCD. Cases that showed mesangial expansion or hypercellularity without any positive IF for immunoglobulins except for IgM were classified as MsPGN-like IgMN.

### 2.2. Study Design and Data Collection

As a prospective observational study, a total of 1791 patients received a native kidney biopsy at eight hospitals (Yeouido St. Mary’s Hospital, Seoul St. Mary’s Hospital, Bucheon St. Mary’s Hospital, Eunpyeong St. Mary’s Hospital, Uijeongbu St. Mary’s Hospital, St. Vincent’s Hospital, Incheon St. Mary’s Hospital, and Daejeon St. Mary’s Hospital) affiliated with The Catholic University of Korea College of Medicine from December 2014 to October 2020. They were registered in the kidney biopsy registry of The Catholic Medical Center with written informed consent. This study was approved by the Institutional Review Board of The Catholic Medical Center (XC19OEDI0025) and the steering committee of the kidney biopsy registry (CMC-KBR-006). It was conducted in accordance with the Declaration of Helsinki. After excluding patients with missing values in data, 94, 57, 81, and 26 patients were diagnosed with IgMN, MCD, FSGS, and MsPGN, respectively, during the period. The average follow-up period was approximately 2 years after kidney biopsy.

Data of clinical information, including laboratory data and pathology reports, were obtained from the kidney biopsy registry. Age, sex, drinking, smoking, co-morbid diseases, systolic blood pressure (SBP), diastolic blood pressure (DBP), and treatment of each patient were collected as demographic and clinical data. Blood biochemistry data included white blood cell (WBC) count, hemoglobin (Hb), hemoglobin A1c (HbA1c), blood urea nitrogen (BUN), creatinine, erythrocyte sedimentation rate (ESR), high-sensitivity C-reactive protein (hs-CRP), total protein, albumin, total cholesterol, ferritin, C3, C4, IgG, IgA, IgM, and IgE levels. Estimated glomerular filtration rate (eGFR) was evaluated according to the Chronic Kidney Disease Epidemiology Collaboration (CKD-EPI) creatinine equation [15,16]. Proteinuria was determined by spot urine protein-to-creatinine ratio (UPCR). These data were collected at the time of kidney biopsy, the 1st visit (10.9 ± 3.3 months), and the 2nd visit (22.5 ± 3.7 months) after kidney biopsy. Renal pathology data included the severity of mesangial matrix expansion, mesangial cell proliferation, crescent and glomerulosclerosis in glomeruli, atrophy, acute tubular necrosis in tubules, and arterial intimal hyalinosis in vessels. These findings were scored (0 (negative), 1 (trace), 2 (mild), 3 (moderate), and 4 (marked)) or calculated as a percentage (%). For immunofluorescence, the extent of IgM, C3, or C4 was calculated as a total sum of deposition scores. The analysis between different LM findings in IgMN was performed by dividing patients into three groups: MCD-like (*n* = 25), FSGS-like (*n* = 21), and MsPGN-like IgMN (*n* = 48).

### 2.3. Statistical Analysis

Continuous variables were presented as mean ± standard deviation (SD) or median (inter quartile range (IQR)). Categorical variables were presented as numbers with percentages. Clinical and pathological comparisons were performed through a Chi-square test with Bonferroni correction for categorical variables and Kruskal-Wallis test with Dwass-Steel-Critchlow-Fligner (DSCF) for continuous variables. The renal outcome during the follow-up in each glomerular disease was analyzed using a mixed model. The fixed or interaction effect included glomerular disease type, follow-up number, or glomerular disease type * follow-up number. The analysis was performed by correcting treatment modalities, such as renin-angiotensin-aldosterone system (RAAS) blockers, furosemide, corticosteroids, rituximab, other immunosuppressants, and plasma exchange. Since the follow-up period was not constant for each patient, a mixed model was performed by including the follow-up period (number of months) as a correction variable in the analysis. In all glomerular disease types, a mixed model analysis was also performed to discover factors related to renal outcome among the remaining variables, including treatment. In addition, a comparative analysis according to different LM morphologies among patients diagnosed with IgMN was performed using a Chi-square test with Bonferroni correction for categorical variables and a Kruskal-Wallis test with DSCF for continuous variables. A mixed model analysis was performed to examine factors related to renal outcome in IgMN. Finally, logistic regression analyses were performed to estimate the odds ratios (ORs) and 95% confidence intervals (CIs) for ≥20% decline of eGFR in IgMN patients. Statistical significance was defined at *p* < 0.05.

## 3. Results

### 3.1. Clinical and Pathological Features of Patients

The prevalence of IgMN in this cohort was 5.2%. (*n* = 94) As shown in Table 1, the mean age at kidney biopsy was older in the FSGS group than in the MsPGN group (*p* = 0.016). However, there was no difference in demographic findings when the IgMN group was compared with other groups. In the IgMN group, three patients and two patients had stable status of hepatitis B and C, respectively. There was no patient with any hematologic disease.

IgMN patients had lower hemoglobin, total cholesterol, C3, and C4 levels than MCD patients (*p* < 0.001, *p* < 0.001, *p* = 0.002, and *p* = 0.013, respectively). However, total protein and albumin concentrations in the IgMN group were higher than those in the MCD group (both *p* < 0.001). Only total cholesterol and serum IgG levels were higher in the IgMN group than in the FSGS group (*p* = 0.040 and *p* = 0.017, respectively). There were significant differences in serum creatinine, eGFR, and proteinuria between IgMN and MCD patients (*p* = 0.019, *p* = 0.002, and *p* = 0.001, respectively). However, there were no significant differences in renal functional parameters between IgMN and MsPGN or FSGS groups.

At baseline, 63.1% of IgMN patients received RAAS blocking therapy. This rate was not significantly different from those of other patients receiving the same therapy (Table 1). Glucocorticoids and furosemide were more frequently used in patients with IgMN than in patients with MCD (*p* < 0.001). However, there was no significant difference in the use of other immunosuppressive agents or RAAS blockades among all groups.

A comparison between patients with IgMN and other glomerular diseases indicated that IgMN had more interstitial fibrosis and tubular atrophy than MCD (*p* = 0.002 and *p* = 0.026, respectively, Table 2). Meanwhile, there was no significant difference in expansion or cell proliferation of the glomerular mesangium among all glomerular diseases.

### 3.2. Renal Outcome

Renal outcomes in patients with glomerular diseases are shown Figure 1 and Table 3. With time effect or group effect only, proteinuria in each group showed a decrease during the follow-up (*p* < 0.001). However, there was no significant interaction between glomerular disease type and follow-up period in any group for change of proteinuria (Figure 2). Serum creatinine and eGFR did not show any change over time among patients with glomerular diseases.

When treatment and follow-up duration were corrected as confusion variables, proteinuria over time in each group was found to be decreased (*p* = 0.006, Table 3). However, there was no significant interaction between glomerular disease type and follow-up period in any group for change of serum creatinine, eGFR, or proteinuria.

Clinical and laboratory indicators related to renal outcome, such as serum creatinine, eGFR, and proteinuria, for patients with all glomerular diseases included age, hypertension, systolic blood pressure, diastolic blood pressure, Hb, HbA1c, BUN, ESR, hs-CRP, ferritin, total protein, albumin, ferritin, C3, C4, IgG, and IgE (Appendix A). Among the pathologic findings, interstitial fibrosis and tubular atrophy at diagnosis were frequently associated with change of serum creatinine, eGFR, or proteinuria. In addition, frequent use of furosemide or glucocorticoids was observed in cases with an increase of proteinuria.

### 3.3. IgMN Subgroup Analysis

When IgMN was divided into MCD-like, FSGS-like, and MsPGN-like IgMN according to the LM finding, there were significant differences in baseline BUN, ESR, total protein, albumin, total cholesterol, C4, and IgG levels among IgMN subgroups (Table 4). Interstitial fibrosis and tubular atrophy were more frequently noted in FSGS-like IgMN than in MCD-like IgMN or MsPGN-like IgMN. FSGS-like IgMN had a higher serum creatinine level but lower eGFR than MsPGN-like IgMN. MsPGN-like IgMN showed less proteinuria than MCD-like or FSGS-like IgMN.

The subtype of IgMN was not a factor that could affect serum creatinine, eGFR, or proteinuria over time (Appendix A). The multivariate logistic regression analysis showed that only the presence of hypertension was an independent risk factor predicting the occurrence of a ≥20% decrease in eGFR, the end-point event, in IgMN patients (Appendix A and Table 5).

## 4. Discussion

Results of this study demonstrated clinicopathological characteristics and the clinical course of the lesser-known IgMN by comparing it with other glomerular diseases, such as MCD, FSGS, and IF-negative MsPGN. We observed that IgMN had a clinical course similar to MCD, FSGS, and IF-negative MsPGN during the follow-up period. We also found that the presence of hypertension at the time of kidney biopsy was an independent risk factor of a decline of eGFR.

Several studies have indicated that IgMN shows a spectrum of pathological changes, ranging from minor changes to FSGS [3,4,17,18]. Considering that transitions from MCD to IgMN and from IgMN to FSGS in subsequent kidney biopsy had been observed in a few previous reports [19,20,21], it could be hypothesized that IgMN, MCD, and FSGS are not separate entities. They might represent a spectrum of diseases that begin with minimal change and end in FSGS [21]. On the contrary, there is a possibility that IgMN, MCD, and FSGS are different and independent entities with overlapping histological appearance [18]. Besides IgMN, glomerular IgM deposition has also been observed in a wide range of secondary renal diseases, including diabetic nephropathy and hypertensive nephropathy, although the significance or pathogenic role of glomerular IgM remains elusive [22]. It is known that IgM in the blood of a normal subject mainly consists of natural polyreactive antibodies. IgMN has been thought to be involved in protection against infection, suppression of autoimmunity, and promotion of wound healing [23,24,25,26]. However, some researchers have shown that IgM may harm in some situations and that IgM could be an important mediator of glomerular injury [22,23]. Experimentally, mice treated with anti-CD20 and reconstituted with IgM purified from kidneys with adriamycin nephropathy, an experimental model of a chemically induced FSGS, showed a trend toward greater proteinuria than mice not reconstituted with IgM [22]. In addition, it has been observed that IgM can activate the complement system in the injured glomeruli [22], suggesting that IgM might be pathogenic, not just as a marker of nonimmune injury [22,23]. In a subsequent study utilizing mice with a targeted deletion of the gene for the complement regulatory protein factor H, a non-sclerotic model of glomerular injury, the alternative complement pathway caused IgM to bind, recruiting the classical pathway to amplify glomerular injury [27]. These findings suggest that binding of IgM within the glomerulus can be a downstream event occurring secondary to glomerular damage [22,27]. However, the involvement of IgM might occur in a subset of patients with any given glomerular lesion [27], since not all have demonstrated a significant role of IgM in nephrotic syndrome [28,29]. Furthermore, the long-term effect of IgM deposition in glomerular diseases are not yet known. When IgMN was classified into three categories (MCD-like, FSGS-like, and MsPGN-like IgMN) based on the LM morphology, there were differences in renal functional parameters, including serum creatinine, eGFR, and proteinuria at the time of diagnosis. Based on our data that clinical and histological findings of IgMN were closer to those of FSGS than to MCD or IF-negative MsPGN over time, the overall clinical course of IgMN might be similar to that of FSGS.

Prognostic factors of this condition varied from study to study. A previous study has shown that hypertension, proteinuria, serum total protein, interstitial fibrosis, and positivity of glomerular C1q are risk factors for end-stage renal disease (ESRD) in IgMN [4]. Another study has demonstrated that renal prognosis of IgMN is associated with glomerulosclerosis or tubular atrophy [3]. In our study, the prevalence of hypertension in IgMN was 33% at the time of kidney biopsy, similar to previously reported ones [4]. Only the presence of hypertension at the time of diagnosis was found to be an independent risk factor for the progression of renal dysfunction in multivariate analysis. However, the presence or absence of IgM in glomerular diseases did not seem to be associated with a greater progression to renal insufficiency when comparing with IgM-negative MCD, FSGS, or MsPGN. Although IgM might be a pathogenic factor involved in a higher degree of glomerular injury as mentioned above, our results did not show that IgMN would have a worse renal prognosis than other IgM-negative glomerular diseases.

This study has some limitations. First, this cohort was a relatively small and ethnically homogeneous population, which might limit the generalizability of the findings. Second, the duration of follow-up was limited. In this study, the longest follow-up period was 27 months. However, no case progressed to ESRD during the study period. It was not possible to evaluate remission due to the relatively short study period and the structural characteristics of the current kidney biopsy registry. Third, with the limited study period and enrolled subject number, it was not possible to analyze response to each treatment in patients in detail. The effectiveness of a specific therapeutic agent, such as rituximab for IgMN, has not been fully investigated yet, although several studies using rituximab as a therapeutic agent for IgMN have been previously reported [30,31]. Since rituximab was used in only one case throughout this cohort, its effect on renal outcome could not be evaluated. Forth, since the biopsies were not analyzed in a perfectly blind fashion, there may be a degree of inter-observer variability. Fifth, IgM clonality was not defined because there were limited data with IgM clonality in the current kidney biopsy registry cohort. Sixth, one of the controls in this study, IF-negative MsPGN, is an expression of histomorphological pattern rather than a disease entity. However, in this study, IF-negative MsPGN was adopted as a control for the purpose of comparing the IgMN patient group with a more diverse control group. Nonetheless, the main strength of this study was the inclusion of biopsy-conformation in patients with a diverse range of kidney diseases, which made it possible to compare with each established glomerular disease. In addition, our study adhered to strict pathological criteria in order to delineate patients with true IgMN. Failure to identify IgM nephropathy as a distinct clinicopathological entity thus far might have resulted from failure to adhere to strict pathological criteria [32]. Finally, we found that IgMN might be a ‘homogenous’ syndrome, since it carried a similar clinical course during the observational period regardless of whether light microscopic findings on the kidney biopsy appeared like MCD, FSGS, or MsPGN.

## 5. Conclusions

In conclusion, we observed that the clinical course and renal prognosis of IgMN was similar to MCD, FSGS, and MsPGN, with clinical and histological findings similar to those of FSGS rather than to MCD or nonspecific MsPGN. We also found that hypertension at presentation was the only significant independent risk factor for declined renal function in IgMN patients. Further research is needed to prove the definite distinctiveness of IgMN and gauge its long-term renal outcome and patient prognosis.

## Figures and Tables

**Figure 1 jcm-10-04191-f001:**
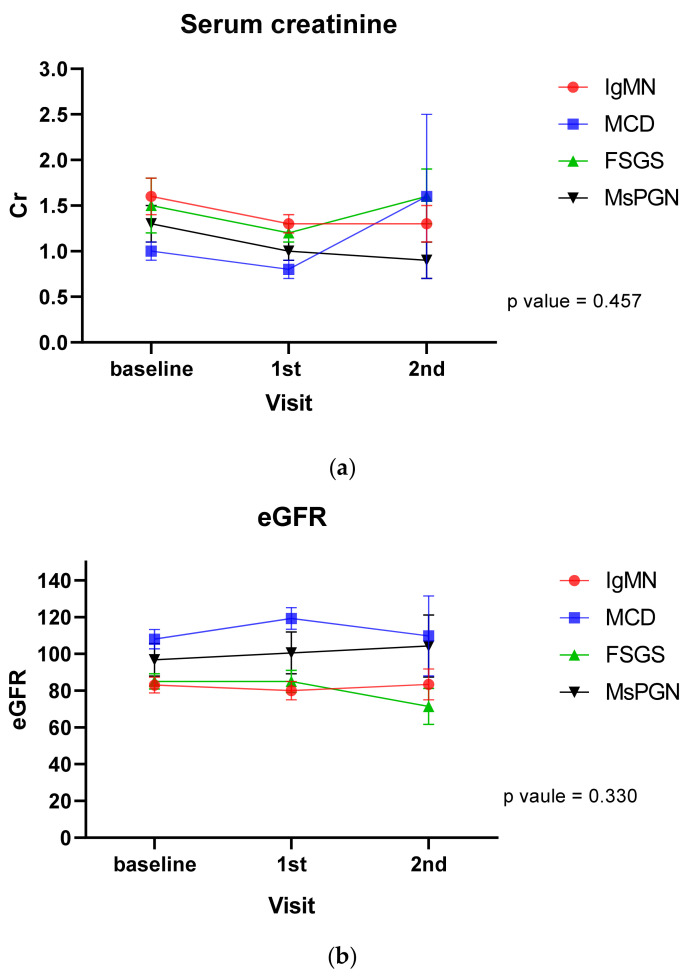
Time course of serum creatinine (**a**), eGFR (**b**) and proteinuria (**c**). eGFR, estimated glomerular filtration rate. The *p* value reflects both the time effect and the group effect. Cr, creatinine; FSGS, focal segmental glomerulosclerosis; IgMN, IgM nephropathy; MCD, minimal change disease; MsPGN, mesangial proliferative glomerulonephritis; UPCR, urine protein-to-creatinine ratio.

**Figure 2 jcm-10-04191-f002:**
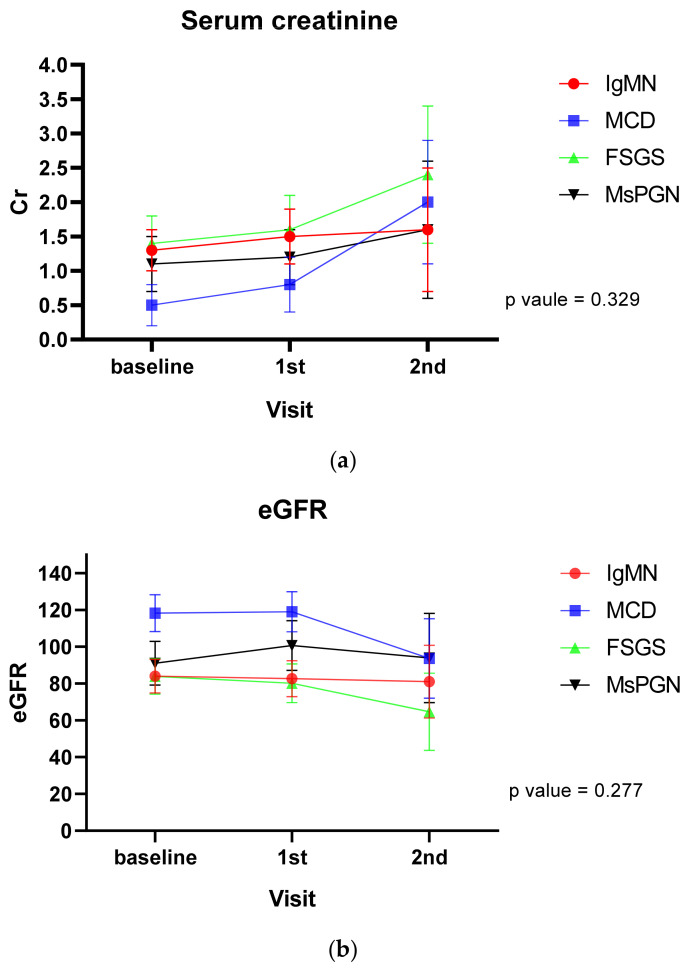
Time course of serum creatinine (**a**), eGFR (**b**), and proteinuria (**c**) corrected by confusion variables. Cr, creatinine; eGFR, estimated glomerular filtration rate. The *p* value reflects both the time effect and the group effect.

**Table 1 jcm-10-04191-t001:** Clinical characteristics of subjects at the time of kidney biopsy.

	Overall (*n* = 258)	IgMN (*n* = 94)	MCD (*n* = 57)	FSGS (*n* = 81)	MsPGN (*n* = 26)	*p* Value	*p* Value for IgMN vs. MCD	*p* Value for IgMN vs. FSGS	*p* Value for IgMN vs. MsPGN	*p* Value for MCD vs. FSGS	*p* Value for MCD vs. MsPGN	*p* Value for FSGS vs. MsPGN
Age, yr	46.5 ± 16.8	46.5 ± 16.1	46.2 ± 18.6	49.5 ± 16.1	38.3 ± 15.9	0.033	0.996	0.675	0.125	0.642	0.234	0.016
Female	123 (47.7)	45 (47.9)	20 (35.1)	44 (54.3)	14 (53.8)	0.142	0.744	>0.999	>0.999	0.154	0.642	>0.999
Drinking	213 (82.6)	79 (84.0)	44 (77.2)	70 (86.4)	20 (76.9)	0.441	>0.999	>0.999	>0.999	0.955	>0.999	>0.999
Smoking						0.095	0.776	>0.999	>0.999	0.044	>0.999	>0.999
Non-smoker	211 (81.8)	77 (81.9)	40 (70.2)	71 (87.7)	23 (88.5)							
Past smoker	13 (5.0)	4 (4.3)	7 (12.3)	1 (1.2)	1 (3.8)							
Current smoker	34 (13.2)	13 (13.8)	10 (17.5)	9 (11.1)	2 (7.7)							
Hypertension	90 (34.9)	31 (33.0)	15 (26.3)	35 (43.2)	9 (34.6)	0.215	>0.999	0.983	>0.999	0.253	>0.999	>0.999
Diabetes	24 (9.3)	7 (7.4)	5 (8.8)	10 (12.3)	2 (7.7)	0.712	0.712	>0.999	>0.999	>0.999	>0.999	>0.999
SBP, mmHg	127.1 ± 16.7	127.6 ± 18.6	125.0 ± 14.6	129.0 ± 15.8	123.2 ± 15.9	0.419	0.933	0.87	0.761	0.568	0.978	0.494
DBP, mmHg	76.5 ± 11.0	76.1 ± 12.2	75.6 ± 11.1	78.1 ± 10.4	75.3 ± 8.2	0.586	0.981	0.548	>0.999	0.909	0.962	0.786
Nephrotic range proteinuria	95 (36.8)	34 (36.2)	37 (64.9)	21 (25.9)	3 (11.5)	<0.001	0.001	0.146	0.016	<0.001	<0.001	0.126
Hematuria	167 (64.7)	58 (61.7)	36 (63.2)	54 (66.7)	19 (73.1)	0.714	0.858	0.495	0.284	0.67	0.375	0.541
WBC, ×10/mm^3^	7.5 ± 2.4	7.8 ± 2.6	7.7 ± 2.9	7.3 ± 1.9	6.9 ± 2.3	0.576	0.947	0.931	0.486	>0.999	0.857	0.748
Hb, g/dL	13.2 ± 2.1	12.8 ± 2.1	14.2 ± 1.6	13.0 ± 2.0	12.8 ± 2.6	<0.001	<0.001	0.998	0.916	0.003	0.162	0.963
HbA1c, %	5.6 ± 0.6	5.6 ± 0.5	5.6 ± 0.6	5.6 ± 0.7	5.4 ± 0.3	0.489	0.972	0.98	0.299	>0.999	0.765	0.703
BUN, mg/dL	21.3 ± 14.2	22.1 ± 14.2	20.0 ± 13.3	22.0 ± 15.5	19.7 ± 11.9	0.45	0.819	0.97	0.706	0.578	0.995	0.603
Creatinine, mg/L	1.4 ±1.7	1.6 ± 1.5	1.0 ± 0.5	1.5 ± 2.5	1.3 ± 1.1	0.032	0.019	0.859	0.787	0.125	0.760	0.958
eGFR, mL/min/1.73 m^2^	90.6 ± 41.6	83.1 ± 41.7	108.0 ± 40.1	85.0 ± 37.9	96.8 ± 45.6	0.001	0.002	0.996	0.443	0.004	0.765	0.440
UPCR	2.0 (0.7–6.2)	1.7 (0.3–6.0)	6.2 (2.0–9.0)	1.8 (0.9–3.9)	0.6 (0.3–1.8)	<0.001	0.001	0.946	0.203	<0.001	<0.001	0.013
ESR, mm/hr	28.4 ± 26.6	26.5 ± 25.6	37.9 ± 29.8	26.6 ± 24.7	21.8 ± 26.0	0.019	0.088	0.967	0.528	0.131	0.047	0.337
hs-CRP, mg/L	0.8 ± 3.4	0.4 ± 1.0	0.8 ± 2.3	0.5 ± 1.6	3.6 ± 9.3	0.908	0.998	0.934	0.997	0.918	0.994	0.994
Total protein, g/dL	6.1 ± 1.4	6.2 ± 1.3	4.9 ± 1.4	6.5 ± 1.1	6.9 ± 0.9	<0001	<0.001	0.482	0.189	<0.001	<0.001	0.762
Albumin, g/dL	3.4 ± 1.1	3.5 ± 1.1	2.4 ± 1.1	3.7 ± 0.9	4.0 ± 0.8	<0001	<0.001	0.74	0.188	<0.001	<0.001	0.462
Total cholesterol, mg/dL	248.4 ± 118.3	234.4 ± 101.8	344.7 ± 133.3	214.8 ± 97.9	184.3 ± 64.5	<0001	<0.001	0.476	0.04	<0.001	<0.001	0.379
Ferritin, ng/mL	192.0 ± 244.1	189.2 ± 230.7	262.2 ± 299.2	157.8 ± 208.5	163.9 ± 264.6	0.031	0.173	0.792	0.899	0.025	0.132	>0.999
C3 depletion	29 (11.7)	15 (16.5)	3 (5.5)	6 (7.7)	5 (21.7)	0.057	0.050	0.084	0.548	0.735	0.045	0.119
C4 depletion	1 (0.4)	0 (0.0)	0 (0.0)	1 (1.3)	0 (0.0)	0.632	N/A	0.462	N/A	<0.999	N/A	>0.999
IgG, mg/dL	940.5 ± 453.7	942.1 ± 392.0	616.9 ± 383.7	1059.0 ± 371.4	1307.4 ± 626.9	<0001	<0.001	0.185	0.017	<0.001	<0.001	0.195
IgA, mg/dL	241.5 ± 115.8	222.2 ± 103.8	264.2 ± 138.6	249.4 ± 117.5	238.1 ± 86.4	0.366	0.346	0.605	0.887	0.923	0.989	>0.999
IgM, mg/dL	162.0 ± 434.0	231.7 ± 702.6	132.9 ± 106.1	115.1 ± 61.1	112.5 ± 57.3	0.88	0.991	0.966	0.985	0.874	0.942	>0.999
IgE, mg/dL	653.9 ± 3078.4	416.3 ± 910.4	1548.6 ± 6377.9	489.3 ± 1451.9	156.8 ± 250.3	0.01	0.242	0.339	0.461	0.018	0.074	0.994
Treatment												
RAAS blocker (%)	152 (63.1)	60 (64.5)	22 (44.0)	56 (73.7)	14 (63.6)	0.009	0.108	>0.999	>0.999	0.005	0.749	>0.999
Furosemide (%)	84 (35.0)	28 (30.1)	37 (74.0)	15 (19.7)	4 (19.0)	<0001	<0001	0.742	>0.999	<0001	<0001	>0.999
Glucocorticoids (%)	106 (44.0)	36 (38.7)	39 (78.0)	23 (30.3)	8 (36.4)	<0001	<0001	>0.999	>0.999	<0001	0.004	>0.999
Rituximab (%)	1 (0.4)	1 (1.1)	0 (0.0)	0 (0.0)	0 (0.0)	>0.999	>0.999	>0.999	>0.999	N/A	N/A	N/A
Other immunosuppressants (%)	10 (4.2)	5 (5.4)	3 (6.0)	2 (2.6)	0 (0.0)	0.653	>0.999	>0.999	>0.999	>0.999	>0.999	>0.999

* Data are expressed as the mean ± SD, n (%), or median (interquartile range (IQR)). The *p* value for the difference between each group was determined by a Chi-square test with Bonferroni correction or a Kruskal-Wallis test with Dwass-Steel-Critchlow-Fligner (DSCF) for multiple comparisons. BUN, blood urea nitrogen; DBP, diastolic blood pressure; ESR, erythrocyte sedimentation rate; FSGS, focal segmental glomerulosclerosis; Hb, hemoglobin; HbA1c, hemoglobin A1c; hs-CRP, high sensitivity c-reactive protein; IgMN, IgM nephropathy; MCD, minimal change disease; MsPGN, mesangial proliferative glomerulonephritis; eGFR, estimated glomerular filtration rate; RAAS, renin-angiotensin-aldosterone system; SBP, systolic blood pressure; UPCR, urine protein-to-creatinine ratio; WBC, white blood cell; Nephrotic range proteinuria, urine protein-to-creatinine ratio > 3.5; Hematuria, Occult blood ≥ ± and RBC > 5 in urinalysis; C3 depletion, <90 mg/dL; C4 depletion, <10 mg/dL.

**Table 2 jcm-10-04191-t002:** Renal pathologic findings of subjects at diagnosis.

	Overall (*n* = 258)	IgMN (*n* = 94)	MCD (*n* = 57)	FSGS (*n* = 81)	MsPGN (*n* = 26)	*p* Value	*p* Value for IgMN vs. MCD	*p* Value for IgMN vs. FSGS	*p* Value for IgMN vs. MsPGN	*p* Value for MCD vs. FSGS	*p* Value for MCD vs. MsPGN	*p* Value for FSGS vs. MsPGN
Glomerulosclerosis, %	22.4 ± 26.0	23.8 ± 26.6	2.5 ± 6.2	33.9 ± 25.4	21.7 ± 28.3	<0001	<0.001	0.011	0.901	<0.001	0.01	0.113
Mesangial matrix expansion						<0001	>0.999	0.35	<0001	0.427	<0001	0.001
0, negative	91 (40.6)	38 (40.4)	22 (50.0)	31 (45.6)	0 (0.0)							
1, trace	72 (32.1)	35 (37.2)	17 (38.6)	16 (23.5)	4 (22.2)							
2, mild	53 (23.7)	18 (19.1)	5 (11.4)	19 (27.9)	11 (61.1)							
3, moderate	5 (2.2)	0 (0.0)	0 (0.0)	2 (2.9)	3 (16.7)							
4, marked	3 (1.3)	3 (3.2)	0 (0.0)	0 (0.0)	0 (0.0)							
Mesangial cell proliferation						<0001	>0.999	0.341	<0001	0.086	<0001	0.001
0, negative	91 (40.6)	38 (40.4)	20 (45.5)	33 (48.5)	0 (0.0)							
1, trace	75 (33.5)	36 (38.3)	20 (45.5)	15 (22.1)	4 (22.2)							
2, mild	53 (23.7)	20 (21.3)	4 (9.1)	18 (26.5)	11 (61.1)							
3, moderate	5 (2.2)	0 (0.0)	0 (0.0)	2 (2.9)	3 (16.7)							
Crescent, %	1.8 ± 11.0	2.4 ± 13.1	0.0 ± 0.0	2.0 ± 12.6	1.7 ± 5.9	0.085	0.396	0.998	0.396	0.341	0.03	0.522
Interstitial fibrosis						0.001	0.002	>0.999	>0.999	<0001	0.016	>0.999
0, negative	78 (34.8)	30 (31.9)	28 (63.6)	16 (23.5)	4 (22.2)							
1, trace	63 (28.1)	23 (24.5)	11 (25.0)	22 (32.4)	7 (38.9)							
2, mild	56 (25.0)	29 (30.9)	5 (11.4)	18 (26.5)	4 (22.2)							
3, moderate	26 (11.6)	11 (11.7)	0 (0.0)	12 (17.6)	3 (16.7)							
4, marked	1 (0.4)	1 (1.1)	0 (0.0)	0 (0.0)	0 (0.0)							
Tubular atrophy						0.003	0.026	>0.999	>0.999	0.002	0.011	>0.999
0, negative	86 (38.4)	34 (36.2)	29 (65.9)	19 (27.9)	4 (22.2)							
1, trace	56 (25.0)	21 (22.3)	9 (20.5)	19 (27.9)	7 (38.9)							
2, mild	56 (25.0)	27 (28.7)	6 (13.6)	19 (27.9)	4 (22.2)							
3, moderate	23 (10.3)	9 (9.6)	0 (0.0)	11 (16.2)	3 (16.7)							
4, marked	3 (1.3)	3 (3.2)	0 (0.0)	0 (0.0)	0 (0.0)							
Acute tubular necrosis						0.529	>0.999	>0.999	>0.999	>0.999	>0.999	>0.999
0, negative	198 (88.4)	84 (89.4)	39 (88.6)	60 (88.2)	15 (83.3)							
1, trace	5 (2.2)	0 (0.0)	2 (4.5)	2 (2.9)	1 (5.6)							
2, mild	11 (4.9)	5 (5.3)	1 (2.3)	4 (5.9)	1 (5.6)							
3, moderate	10 (4.5)	5 (5.3)	2 (4.5)	2 (2.9)	1 (5.6)							
Arterial intimal hyalinosis						0.077	>0.999	0.326	0.125	>0.999	>0.999	>0.999
0, negative	194 (87.8)	85 (91.4)	38 (90.5)	57 (83.8)	14 (77.8)							
1, trace	15 (6.8)	7 (7.5)	3 (7.1)	4 (5.9)	1 (5.6)							
2, mild	8 (3.6)	1 (1.1)	1 (2.4)	5 (7.4)	1 (5.6)							
3, moderate	4 (1.8)	0 (0.0)	0 (0.0)	2 (2.9)	2 (11.1)							
Fibrous wall thickening						0.869	>0.999	>0.999	>0.999	>0.999	>0.999	>0.999
0, negative	148 (67.0)	62 (66.7)	31 (73.8)	44 (64.7)	11 (61.1)							
1, trace	19 (8.6)	7 (7.5)	4 (9.5)	7 (10.3)	1 (5.6)							
2, mild	29 (13.1)	15 (16.1)	4 (9.5)	7 (10.3)	3 (16.7)							
3, moderate	22 (10.0)	8 (8.6)	3 (7.1)	9 (13.2)	2 (11.1)							
4, marked	3 (1.4)	1 (1.1)	0 (0.0)	1 (1.5)	1 (5.6)							
Foot process effacement						<0.001	<0.001	<0.001	0.589	0.163	<0.001	0.003
0, no	38 (16.5)	20 (21.7)	5 (9.6)	10 (14.5)	3 (17.6)							
1, focal	42 (18.3)	28 (30.4)	1 (1.9)	1 (7.2)	8 (47.1)							
2, diffuse	150 (65.2)	44 (47.8)	46 (88.5)	2 (78.3)	6 (35.3)							

* Data are expressed as the mean ± SD or n (%). The *p* value for the difference between each group was determined by a Chi-square test with Bonferroni correction or Kruskal–Wallis test with Dwass-Steel-Critchlow-Fligner (DSCF) for multiple comparisons. FSGS, focal segmental glomerulosclerosis; IgMN, IgM nephropathy; MCD, minimal change disease; MsPGN, mesangial proliferative glomerulonephritis.

**Table 3 jcm-10-04191-t003:** Renal outcome during the follow-up after kidney biopsy.

	IgMN	MCD	FSGS	MsPGN	*p* Value (Time Effect)	*p* Value (Group Effect)	*p* Value (Time*Group Effect)	*p* Value for IgMN vs. MCD	*p* Value for IgMN vs. FSGS	*p* Value for IgMN vs. MsPGN	*p* Value for MCD vs. FSGS	*p* Value for MCD vs. MsPGN	*p* Value for FSGS vs. MsPGN
Serum creatinine					0.232	0.141	0.329						
Baseline	1.3 ± 0.3	0.5 ± 0.3	1.4 ± 0.4	1.1 ± 0.4				<0.001	0.867	0.376	0.005	0.011	0.444
1st visit	1.5 ± 0.4	0.8 ± 0.4	1.6 ± 0.5	1.2 ± 0.4				0.02	0.783	0.488	0.054	0.287	0.433
2nd visit	1.6 ± 0.9	2.0 ± 0.9	2.4 ± 1.0	1.6 ± 1.0				0.446	0.132	0.964	0.492	0.492	0.178
eGFR					0.113	0.001	0.277						
Baseline	84.1 ± 9.2	118.3 ± 10.0	83.9 ± 9.7	91.1 ± 11.9				<0.001	0.98	0.44	<0.001	0.011	0.439
1st visit	82.7 ± 9.8	119.0 ± 10.9	80.2 ± 10.5	100.7 ± 13.5				<0.001	0.745	0.125	<0.001	0.161	0.087
2nd visit	81.0 ± 19.8	93.7 ± 21.6	64.7 ± 21.0	93.9 ± 24.3				0.331	0.119	0.377	0.037	0.99	0.053
UPCR					0.006	0.522	0.062						
Baseline	4.1 ± 1.0	5.2 ± 1.1	3.5 ± 1.0	2.4 ± 1.2				0.173	0.381	0.082	0.035	0.008	0.225
1st visit	3.3 ± 1.0	1.4 ± 1.2	3.5 ± 1.0	3.0 ± 1.3				0.107	0.831	0.835	0.058	0.264	0.704
2nd visit	5.1 ± 2.3	4.8 ± 2.6	6.1 ± 2.3	5.0 ± 2.7				0.8896	0.4184	0.9501	0.461	0.948	0.5023

* Data are expressed as the mean ± SD or median (interquartile range (IQR)). Mixed effects models with time (baseline, 1st visit, 2nd visit), group (glomerular disease type), or time*group as fixed effects were applied with random intercepts and slopes for each individual value, allowing unstructured correlation between the random effects. Pairwise comparisons were adjusted by using Bonferroni’s procedure to account for multiple testing. FSGS, focal segmental glomerulosclerosis; IgMN, IgM nephropathy; MCD, minimal change disease; MsPGN, mesangial proliferative glomerulonephritis; UPCR, urine protein-to-creatinine ratio.

**Table 4 jcm-10-04191-t004:** Clinical and pathological comparisons among IgMN subtypes according to light microscopic finding.

	Overall (*n* = 94)	MCD-Like (*n* = 25)	FSGS-Like (*n* = 21)	MsPGN (*n* = 48)	*p* Value	*p* Value for MCD-Like vs. FSGS-Like	*p* Value for MCD-Like vs. MsPGN-Like	*p* Value for FSGS-Like vs. MsPGN-Like
Clinical findings								
Age	46.5 ± 16.1	45.7 ± 16.5	50.9 ± 19.3	44.9 ± 14.3	0.311	0.498	0.986	0.285
Female	45 (47.9)	12 (48.0)	9 (42.9)	24 (50.0)	0.861	>0.999	>0.999	>0.999
Drinking	15 (16.0)	5 (20.0)	4 (19.0)	6 (12.5)	0.614	>0.999	>0.999	>0.999
Smoking					0.056	>0.999	>0.999	0.030
Non-smoker	77 (81.9)	20 (80.0)	14 (66.7)	43 (89.6)				
Past smoker	4 (4.3)	1 (4.0)	3 (14.3)	0 (0.0)				
Current smoker	13 (13.8)	4 (16.0)	4 (19.0)	5 (10.4)				
Hypertension	31 (33.0)	6 (24.0)	11 (52.4)	14 (29.2)	0.091	0.141	>0.999	0.195
Diabetes	7 (7.4)	1 (4.0)	3 (14.3)	3 (6.3)	0.501	0.954	>0.999	>0.999
SBP, mmHg	127.6 ± 18.6	124.2 ± 15.5	135.9 ± 25.1	125.8 ± 15.8	0.122	0.164	0.877	0.169
DBP, mmHg	76.1 ± 12.2	76.8 ± 8.9	76.8 ± 19.7	75.3 ± 9.3	0.795	0.890	0.796	0.954
Nephrotic range proteinuria	34 (36.2)	25 (100)	9 (42.9)	0 (0)	<0.001	<0.001	<0.001	<0.001
Hematuria	58 (61.7)	17 (68)	12 (57.1)	29 (60.4)	0.729	0.452	0.527	0.800
WBC, ×10/mm^3^	7.8 ± 2.6	8.2 ± 3.0	8.5 ± 2.4	7.3 ± 2.4	0.087	0.610	0.427	0.091
Hb, g/dL	12.8 ± 2.1	12.7 ± 2.6	13.2 ± 2.1	12.7 ± 1.9	0.824	0.976	0.935	0.817
HbA1c, %	5.6 ± 0.5	5.5 ± 0.6	5.9 ± 0.6	5.5 ± 0.4	0.084	0.164	0.985	0.091
BUN, mg/dL	22.1 ± 14.2	25.4 ± 14.7	26.5 ± 17.1	18.4 ± 11.6	0.015	0.998	0.083	0.028
ESR, mm/hr	26.5 ± 25.6	45.8 ± 25.7	26.1 ± 27.2	16.5 ± 18.6	<0.001	0.022	<0.0001	0.130
hs-CRP, mg/L	0.4 ± 1.0	0.6 ± 1.3	0.4 ± 0.6	0.4 ± 1.0	0.592	0.753	0.929	0.594
Total protein, g/dL	6.2 ± 1.3	5.1 ± 1.1	5.9 ± 1.4	6.9 ± 0.8	<0.001	0.118	<0.0001	0.016
Albumin, g/dL	3.5 ± 1.1	2.4 ± 0.8	3.2 ± 1.1	4.2 ± 0.6	<0.001	0.019	<0.0001	0.001
Total cholesterol, mg/dL	234.4 ± 101.8	325.9 ± 108.2	226.5 ± 84.3	190.1 ± 70.4	<0.001	0.009	<0.0001	0.158
ferritin, ng/mL	189.2 ± 230.7	236.4 ± 328.1	213.7 ± 159.7	152.3 ± 181.6	0.165	0.595	0.541	0.177
C3 depletion	15 (16.5)	3 (12.5)	2 (9.5)	10 (21.7)	0.383	0.754	0.349	0.230
C4 depletion	0 (0.0)	0 (0.0)	0 (0.0)	0 (0.0)	>0.999	>0.999	>0.999	>0.999
IgG, mg/dL	942.1 ± 392.0	776.1 ± 476.0	935.7 ± 353.3	1031.7 ± 336.3	0.011	0.170	0.009	0.560
IgA, mg/dL	222.2 ± 103.8	234.4 ± 124.4	241.4 ± 67.3	207.0 ± 105.5	0.288	0.907	0.654	0.260
IgM (serum), mg/dL	231.7 ± 702.6	331.7 ± 786.1	127.3 ± 75.0	229.3 ± 813.3	0.262	0.901	0.272	0.572
Ig E (serum), mg/dL	416.3 ± 910.4	663.8 ± 1590.1	694.7 ± 779.3	159.6 ± 240.3	0.077	0.448	0.793	0.053
Creatinine, mg/dL, baseline	1.6 ± 1.5	1.7 ± 1.5	2.0 ± 1.5	1.3 ± 1.6	0.001	0.296	0.208	<0.001
eGFR, baseline	83.1 ± 41.7	77.8 ± 46.0	56.3 ± 32.6	97.6 ± 36.8	<0.001	0.276	0.183	<0.001
UPCR, baseline	1.7 (0.3–6.0)	6.3 (5.6–10.4)	2.1 (1.5–7.3)	0.4 (0.1–1.3)	<0.001	0.089	<0.001	<0.001
Pathologic findings								
Glomerulosclerosis, %	23.8 ± 26.6	19.5 ± 26.8	35.3 ± 27.2	21.0 ± 25.3	0.023	0.039	0.883	0.040
Mesangial matrix expansion					0.598	0.977	>0.999	>0.999
0, negative	38 (40.4)	11 (44.0)	7 (33.3)	20 (41.7)				
1, trace	35 (37.2)	7 (28.0)	11 (52.4)	17 (35.4)				
2, mild	18 (19.1)	5 (20.0)	3 (14.3)	10 (20.8)				
3, moderate	0 (0.0)	0 (0.0)	0 (0.0)	0 (0.0)				
4, marked	3 (3.2)	2 (8.0)	0 (0.0)	1 (2.1)				
Mesangial cell proliferation					0.987	>0.999	>0.999	>0.999
0, negative	38 (40.4)	10 (40.0)	8 (38.1)	20 (41.7)				
1, trace	36 (38.3)	9 (36.0)	9 (42.9)	18 (37.5)				
2, mild	20 (21.3)	6 (24.0)	4 (19.0)	10 (20.8)				
3, moderate	0 (0.0)	0 (0.0)	0 (0.0)	0 (0.0)				
Crescent, %	2.4 ± 13.1	3.6 ± 17.3	0.7 ± 2.3	2.6 ± 13.8	0.688	0.810	0.999	0.700
Interstitial fibrosis					0.002	0.007	0.38	0.005
0, negative	30 (31.9)	9 (36.0)	3 (14.3)	18 (37.5)				
1, trace	23 (24.5)	8 (32.0)	1 (4.8)	14 (29.2)				
2, mild	29 (30.9)	3 (12.0)	12 (57.1)	14 (29.2)				
3, moderate	11 (11.7)	4 (16.0)	5 (23.8)	2 (4.2)				
4, marked	1 (1.1)	1 (4.0)	0 (0.0)	0 (0.0)				
Tubular atrophy					0.001	0.005	0.253	0.008
0, negative	34 (36.2)	11 (44.0)	4 (19.0)	19 (39.6)				
1, trace	21 (22.3)	6 (24.0)	1 (4.8)	14 (29.2)				
2, mild	27 (28.7)	3 (12.0)	11 (52.4)	13 (27.1)				
3, moderate	9 (9.6)	2 (8.0)	5 (23.8)	2 (4.2)				
4, marked	3 (3.2)	3 (12.0)	0 (0.0)	0 (0.0)				
Acute tubular necrosis					0.815	>0.999	>0.999	>0.999
0, negative	84 (89.4)	22 (88.0)	18 (85.7)	44 (91.7)				
1, trace	0 (0.0)	0 (0.0)	0 (0.0)	0 (0.0)				
2, mild	5 (5.3)	1 (4.0)	2 (9.5)	2 (4.2)				
3, moderate	5 (5.3)	2 (8.0)	1 (4.8)	2 (4.2)				
Arterial intimal hyalinosis					0.182	0.614	0.267	>0.999
0, negative	85 (91.4)	24 (96.0)	19 (90.5)	42 (89.4)				
1, trace	7 (7.5)	0 (0.0)	2 (9.5)	5 (10.6)				
2, mild	1 (1.1)	1 (4.0)	0 (0.0)	0 (0.0)				
3, moderate	0 (0.0)	0 (0.0)	0 (0.0)	0 (0.0)				
Fibrous wall thickening					0.178	0.387	0.404	>0.999
0, negative	62 (66.7)	15 (60.0)	17 (81.0)	30 (63.8)				
1, trace	7 (7.5)	0 (0.0)	1 (4.8)	6 (12.8)				
2, mild	15 (16.1)	7 (28.0)	1 (4.8)	7 (14.9)				
3, moderate	8 (8.6)	2 (8.0)	2 (9.5)	4 (8.5)				
4, marked	1 (1.1)	1 (4.0)	0 (0.0)	0 (0.0)				
Foot process effacement					0.002	0.147	0.055	<0.001
0, no	20 (21.7)	5 (20.8)	3 (14.3)	12 (25.5)				
1, focal	28 (30.4)	5 (20.8)	1 (4.8)	22 (46.8)				
2, diffuse	44 (47.8)	14 (58.3)	17 (81.0)	13 (27.7)				

* Data are expressed as the mean ± SD, n (%), or median (interquartile range (IQR)). The *p* value for the difference between each group was determined by a Chi-square test with Bonferroni correction or Kruskal-Wallis test with Dwass-Steel-Critchlow-Fligner (DSCF) for multiple comparisons. BUN, blood urea nitrogen; DBP, diastolic blood pressure; ESR, erythrocyte sedimentation rate; FSGS, focal segmental glomerulosclerosis; Hb, hemoglobin; HbA1c, hemoglobin A1c; hs-CRP, high sensitivity c-reactive protein; IgMN, IgM nephropathy; MCD, minimal change disease; MsPGN, mesangial proliferative glomerulonephritis; SBP, systolic blood pressure; UPCR, urine protein-to-creatinine ratio; WBC, white blood cell; Nephrotic range proteinuria, urine protein-to-creatinine ratio > 3.5; Hematuria, Occult blood ≥ ± and RBC > 5 in urinalysis; C3 depletion, <90 mg/dL; C4 depletion, <10 mg/dL

**Table 5 jcm-10-04191-t005:** Multivariate logistic regression for ≥20% decline in eGFR over time in IgMN patients.

	Adjusted OR (95% Cl)	*p* Value
Hypertension		
No	1	
Yes	4.52 (1.07–19.05)	0.04
Immunosuppressant		
No use	1	
Use	7.83 (1.00–61.51)	0.051

* OR, odds ratio.

## Data Availability

All data are reported in the article.

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
