# Peer review of "Renal Outcome of IgM Nephropathy: A Comparative Prospective Cohort Study"

_jcm, 2021, doi:10.3390/jcm10184191_

Round 1

Reviewer 1 Report

The manuscript by Chae et al "Renal Outcome of IgM Nephropathy: A Comparative Prospective Cohort Study" is based on data from a large number of renal biopsies from several hospitals and ultimately includes over 90 patients with IgMN. However, the criteria for defining IgMN appear to be loosely covered. 
Indeed, IgM nephropathy is an idiopathic glomerular disease characterized by definition by diffuse IgM deposition by immunofluorescence with an intensity of at least 2++ in non-sclerotic glomeruli. It is a controversial entity in which IgM-mediated injury is likely a secondary phenomenon triggered after a primary podocyte insult. IgM could subsequently activate complement. IgMN is also characterized by diffuse podocyte effacement; therefore, electron microscopy is required for the diagnosis of IgM nephropathy. Many studies have shown that children with MCD may show IgM deposition in immunofluorescence, which do not influence the responsivenes to steroids and prognosis of renal disease.
In addition, electron microscopy (EM) is critical for the diagnosis of MCD (and FSGS), but this study does not provide data regarding EM results. There are no data regarding primary, genetic or secondary FSGS with different prognosis and clinical course.
Regarding the design, the study includes a heterogeneous group of diseases used for comparison, with a relatively short follow-up period (at least in terms of defining interesting final clinical goals such as remission, relapse and transition to dialysis). The treatment approach, which is the basic requirement for a scientifically reliable comparison of clinical outcomes (which are hardly comparable under different treatment regimens!) in the different groups, was not investigated (and was probably subordinate to the individual protocol of each clinic). Such a "rare" comparison is therefore of very questionable scientific validity.
However, the article attempts to find the distinguishing features of the IgMN disease entity, which is not yet recognized as isolated by much of the professional community, and in this sense I support efforts to define IgMN more precisely and to distinguish its course from similar clinicopathological entities.

Important questions need to be clarified and important refinements introduced:
1. how many pathologists and under what conditions (double-blind method?) did they review the renal biopsy specimens? This should be specified precisely! There are no data on the intensity of IgM deposits by immunofluorescence. IgM mesangial deposits should be defined as having an intensity of at least 2+ on a semiquantitative score of 0 to 3+
2. electron microscopic data is very important for diagnosis and missing.
3. since it is a disease with IgM deposits often associated with certain diseases (inflammatory, hematologic, ..), it is necessary to provide clinical data on the presence of chronic infections (e.g. HCV, HBV,..) and hematologic diseases (e.g. MGUS. Waldenstroem's disease).
4. it must be clearly stated whether IgM clonality is defined by kappa and lambda light chains 
5. for basic clinical data, I recommend presenting the initial clinical presentation of the disease in each group (how many patients had nephrotic syndrome, acute nephritic syndrome, chronic nephritic syndrome, minimal urine abnormalities, gross hematuria, RPGN) and not just absolute values of albumin, cholesterol, etc.
6. data on complement fractions C3 and C4 may be presented and compared as attributive variables rather than absolute numerical values. It must be stated whether the patient had normal (in laboratory reference values) or reduced complement values and then compare the proportions of patients with reduced complement values 
7. Clinical follow-up should be represented not only by monitoring eGF and UPCR, but especially by determining whether clinical and laboratory remission of the disease has been achieved and if so, whether it is partial or complete 
8. It is almost necessary to provide more detailed data on steroid treatment: how many patients in each group are steroid dependent, steroid resistant and steroid responsive! Are there differences between IgMN and other entities in terms of steroid response?
9. IF -neg MsPGN is only a histomorphological pattern of glomerular damage and not a disease entity. It could encompass several diseases, including early diabetic glomerulosclerosis, tubulointerstitial or vascular diseases, and should be further specified etiopatogenetically in the study 
10. Mesangial proliferation and IF /TA should be assessed according to more recognized schemes, such as the Oxford classification 
11. There is a discrepancy between the sentence Discussion in line 241 "We observed that ......" and in Conclusion in line 317 "In conclusion, ...

Author Response

Sep 08, 2021

Ms No jcm-1367151 “Renal Outcome of IgM Nephropathy: A Comparative Prospective Cohort Study”

Responses to editor and reviewers' comments

Thank you for giving us an opportunity to revise our manuscript. We believe that the comments by the reviewers have improved our paper greatly. Please understand that we did our best to answer the questions and take in the reviewers’ advices. We have denoted the changes in the text marked in red color. Responses to reviewers are outlined below.

Reviewer #1

The manuscript by Chae et al "Renal Outcome of IgM Nephropathy: A Comparative Prospective Cohort Study" is based on data from a large number of renal biopsies from several hospitals and ultimately includes over 90 patients with IgMN. However, the criteria for defining IgMN appear to be loosely covered. Indeed, IgM nephropathy is an idiopathic glomerular disease characterized by definition by diffuse IgM deposition by immunofluorescence with an intensity of at least 2++ in non-sclerotic glomeruli. It is a controversial entity in which IgM-mediated injury is likely a secondary phenomenon triggered after a primary podocyte insult. IgM could subsequently activate complement. IgMN is also characterized by diffuse podocyte effacement; therefore, electron microscopy is required for the diagnosis of IgM nephropathy. Many studies have shown that children with MCD may show IgM deposition in immunofluorescence, which do not influence the responsiveness to steroids and prognosis of renal disease. In addition, electron microscopy (EM) is critical for the diagnosis of MCD (and FSGS), but this study does not provide data regarding EM results. There are no data regarding primary, genetic or secondary FSGS with different prognosis and clinical course. Regarding the design, the study includes a heterogeneous group of diseases used for comparison, with a relatively short follow-up period (at least in terms of defining interesting final clinical goals such as remission, relapse and transition to dialysis). The treatment approach, which is the basic requirement for a scientifically reliable comparison of clinical outcomes (which are hardly comparable under different treatment regimens!) in the different groups, was not investigated (and was probably subordinate to the individual protocol of each clinic). Such a "rare" comparison is therefore of very questionable scientific validity. However, the article attempts to find the distinguishing features of the IgMN disease entity, which is not yet recognized as isolated by much of the professional community, and in this sense I support efforts to define IgMN more precisely and to distinguish its course from similar clinicopathological entities. Important questions need to be clarified and important refinements introduced:

1-1. How many pathologists and under what conditions (double-blind method?) did they review the renal biopsy specimens? This should be specified precisely!

Response: Thank you for reviewing our manuscript and for your important comments. The renal biopsy specimens were usually reviewed by two general pathologists and then by a pathologist with expertise in renal pathology in each institution, who provided semiquantitative scores of glomerular, tubular and vascular injury. All affiliated hospitals share a uniform report form of renal biopsy with quantified histologic grading.  Disagreements between the pathologists on the interpretation were solved with discussion and inter-institutional pathology consultation in which they had to re-review the biopsies and come to agreement. Since the biopsies were not analyzed in a perfectly blind fashion, there may be a degree of inter-observer variability. This weakness was added in addition to limitations (page 14)

1-2. There are no data on the intensity of IgM deposits by immunofluorescence. IgM mesangial deposits should be defined as having an intensity of at least 2+ on a semiquantitative score of 0 to 3+

Response: We entirely agree with you. Previous studies have showed variations in the definition of IgMN: for example, Connor TM, et al. [Nephrol Dial Transplant. 2017;32:823-829] adopted three criteria to meet the definition of IgMN including the biopsy finding that the intensity of IgM staining (graded on a semi-quantitative scale) should be more than trace. For this study, we defined the IgMN as follows: first, we excluded cases with positivity of other immunoglobulins except for IgM. Second, there must be dominant staining for IgM in glomeruli by IF, which intensity should be +1 or higher. To clarify the study design and avoid misunderstandings, the following has been added to the definition section of Materials and Methods (page 2): “Ig M positivity 1+ to 3+”.

  1. Electron microscopic data is very important for diagnosis and missing.

Response: Thank you for your valuable comments. IgMN is an idiopathic glomerular disease characterized by diffuse deposits of IgM in glomeruli at IF. Although there is still no consensus for the diagnostic criteria of IgMN with respect to the intensity of IgM staining or electron microscopic (EM) findings [Connor TM, et al. Nephrol Dial Transplant. 2017;32:823-829], one of criteria to meet the definition of IgMN is that There had to be definite mesangial deposits on EM. We confirmed the presence of mesangial dense deposits on EM by its definition. Furthermore, in order to characterize the disease entity of IgM nephropathy more concretely, we have assessed the presence of foot process effacement in EM findings in each groups and added the results in Table 2 and 4.

  1. Since it is a disease with IgM deposits often associated with certain diseases (inflammatory, hematologic, ..), it is necessary to provide clinical data on the presence of chronic infections (e.g. HCV, HBV,..) and hematologic diseases (e.g. MGUS. Waldenstroem's disease).

Response: Thank you for your suggestions. As per the reviewer’s recommendation, we added the related clinical data on in the ‘Results’ section: “In the IgMN group, three patients and two patients had stable status of hepatitis B and C, respectively. There was no patient with any hematologic disease.” (page 4).

  1. It must be clearly stated whether IgM clonality is defined by kappa and lambda light chains 
    Response: We thank the reviewer for pointing this out. Unfortunately, there were limited data with IgM clonality in the current GN registry cohort, and we added the limitations in ‘Discussion’ section. (page 14)

  1. For basic clinical data, I recommend presenting the initial clinical presentation of the disease in each group (how many patients had nephrotic syndrome, acute nephritic syndrome, chronic nephritic syndrome, minimal urine abnormalities, gross hematuria, RPGN) and not just absolute values of albumin, cholesterol, etc.

Response: As suggested by the reviewers, the description of the clinical presentation can be helpful in understanding each glomerular disease. However, it was difficult to specify the initial clinical symptoms other than proteinuria or hematuria prior to biopsy with the current cohort data. We added the proportion of patients with proteinuria of the nephrotic range and those with hematuria in Table 1.

  1. Data on complement fractions C3 and C4 may be presented and compared as attributive variables rather than absolute numerical values. It must be stated whether the patient had normal (in laboratory reference values) or reduced complement values and then compare the proportions of patients with reduced complement values 

Response: We agree completely with the reviewer’s opinion. As per the reviewer’s recommendation, we edited the presentation of C3 and C4 data ​​as the number (proportion) of patients with complement depletion, instead of the absolute values (Table 1 and Table 4).

  1. Clinical follow-up should be represented not only by monitoring eGFR and UPCR, but especially by determining whether clinical and laboratory remission of the disease has been achieved and if so, whether it is partial or complete 

Response: We thank the reviewer for insightful comments. However, it was not possible to evaluate remission due to the relatively short study period and the structural characteristics of the current kidney biopsy registry. We have added this limitation in the ‘Discussion’ section (page 14).

  1. It is almost necessary to provide more detailed data on steroid treatment: how many patients in each group are steroid dependent, steroid resistant and steroid responsive! Are there differences between IgMN and other entities in terms of steroid response?

Response: Thank you for your valuable comments. However, with the limited study period and enrolled subject number, it was not possible to analyze response to each treatment in patients in detail. Furthermore, it was difficult to evaluate the steroid responsiveness because the timing and duration of steroid treatment, compliance, and the type of steroid used were diverse from patient to patient. Instead, when we examined the factors related to renal outcome of IgMN, we adjusted for whether the patient had initiated steroids treatment. The related limitation was added in ‘Discussion’ section (page 14)

  1. IF-neg MsPGN is only a histomorphological pattern of glomerular damage and not a disease entity. It could encompass several diseases, including early diabetic glomerulosclerosis, tubulointerstitial or vascular diseases, and should be further specified etiopatogenetically in the study 

Response: We agree completely with the reviewer’s opinion. In this study, the MsPGN group was included for the purpose of comparing IgMN with more diverse patient groups, but it is true that IF-neg MsPGN cannot be regarded as a disease entity. The above has been added as follows: “One of the controls in this study, IF-negative MsPGN, is an expression of histomorphological pattern rather than a disease entity. However, in this study, IF-negative MsPGN was adopted as a control for the purpose of comparing the IgMN patient group with a more diverse control group.” (page 14)

  1. Mesangial proliferation and IF /TA should be assessed according to more recognized schemes, such as the Oxford classification 
    Response: Thank you for your important comments. As you said, it would have been better if it was marked with a more recognized scheme such as Oxford classification. However, since the histological grading for all kidney biopsies had been assessed on the widely used scoring system by pathologists, we adopted the pathological variables of the original biopsy report unchanged. Furthermore, as we know, the Oxford classification was developed as a pathological system for IgA nephropathy, and we have no idea that the Oxford classification is available to apply at other glomerular diseases.

  1. There is a discrepancy between the sentence Discussion in line 241 "We observed that ......" and in Conclusion in line 317 "In conclusion, ...

Response: Thank you for your helpful comments. In order to express the intention more clearly, the sentence of 317 has been modified as follows: “In conclusion, we observed that the clinical course and renal prognosis of IgMN similar to MCD, FSGS, and MsPGN with clinical and histological findings similar to those of FSGS rather than to MCD or nonspecific MsPGN.” (page 14)

All reviewer comments were helpful and they provided new insight into our work. We have carefully reviewed and revised the entire manuscript, in addition to the recommended changes. Thank you in advance for your consideration of the revised manuscript.

Yura Chae, Hye Eun Yoon, Yoon Kyung Chang, Young Soo Kim, Hyung Wook Kim, Bum Soon Choi, Cheol Whee Park, Ho Cheol Song, Young Ok Kim, Eun Sil Koh and Sungjin Chung

Reviewer 2 Report

This is a thorough study of clinical and histopathological features of IgM nephropathy in a good sized cohort of patients. The main strength is the detail of the clinical observations. The conclusions are appropriate. I have only minor suggestions for improvement.

Line 76: should be "complement", not "complementary"

Line 156: give the number of patients with IgMN as well as the percentage.

Table 1: the numbers for IgG, IgA, IgM and IgE are too close together - cannot be read.

Figures 1 and 2: consider including p-values for Kruskal-Wallis test in chart.

Author Response

Sep 08, 2021

Ms No jcm-1367151 “Renal Outcome of IgM Nephropathy: A Comparative Prospective Cohort Study”

Responses to editor and reviewers' comments

Thank you for giving us an opportunity to revise our manuscript. We believe that the comments by the reviewers have improved our paper greatly. Please understand that we did our best to answer the questions and take in the reviewers’ advices. We have denoted the changes in the text marked in red color. Responses to reviewers are outlined below.

Reviewer #2

This is a thorough study of clinical and histopathological features of IgM nephropathy in a good sized cohort of patients. The main strength is the detail of the clinical observations. The conclusions are appropriate. I have only minor suggestions for improvement.

  1. Line 76: should be "complement", not "complementary"

Response: Thank you for reviewing our manuscript and for your insightful comments.  As per the reviewer’s comment. We corrected the word as advised. Thank you.

  1. Line 156: give the number of patients with IgMN as well as the percentage.

Response: As per the reviewer’s recommendation, we marked the number of patients.

  1. Table 1: the numbers for IgG, IgA, IgM and IgE are too close together - cannot be read.

Response: As the reviewer mentioned, it was hard to make out the numbers because they are large numbers. Accordingly, we reduced font size of Table 1 to fix it.

  1. Figures 1 and 2: consider including p-values for Kruskal-Wallis test in chart.

Response:  Thank you for your comments. As the reviewer pointed out, we have included p-values ​​in figures 1 and 2.

All reviewer comments were helpful and they provided new insight into our work. We have carefully reviewed and revised the entire manuscript, in addition to the recommended changes. Thank you in advance for your consideration of the revised manuscript.

Yura Chae, Hye Eun Yoon, Yoon Kyung Chang, Young Soo Kim, Hyung Wook Kim, Bum Soon Choi, Cheol Whee Park, Ho Cheol Song, Young Ok Kim, Eun Sil Koh and Sungjin Chung

Round 2

Reviewer 1 Report

Given that the authors took into account all the comments and improved the article in accordance with the data available from the register (including the explanation of the study limitations), I support the publication of the amended article.

Best regards,

Željka Večerić-Haler